# The Role of Courtyards within Acute Mental Health Wards: Designing with Recovery in Mind

**DOI:** 10.3390/ijerph191811414

**Published:** 2022-09-10

**Authors:** Jacqueline McIntosh, Bruno Marques, Gabrielle Jenkin

**Affiliations:** 1Wellington School of Architecture, Victoria University of Wellington, P.O. Box 600, Wellington 6140, New Zealand; 2Department of Psychological Medicine, University of Otago, Wellington 23a Mein St., Wellington 6021, New Zealand

**Keywords:** courtyard design, acute mental health, therapeutic environments, confinement, incarceration, mental health and wellbeing

## Abstract

The role of courtyards and other outdoor spaces in the recovery of acute mental healthcare users has been gaining international appreciation and recognition. However, the physical properties and conditions necessary for therapeutic and rehabilitative engagement remain to be clearly established. This paper contributes to that knowledge by triangulating evidence from the literature, exemplar case studies of good practice and first-hand accounts of the experiences of staff and service users from four acute mental health facilities. The findings are then aligned with a well-established recovery framework (CHIMES) in light of existing landscape architecture knowledge. Within the complexity of varied mental health environments, this work establishes landscape architectural design requirements and qualities essential for recovery. Rather than adopting a prescriptive quantitative approach setting out areas, numbers of elements, etc., the proposed framework recommends a performance-based model and the creation of a cohesive network of microspaces that mesh into a design of outdoor areas. In this way, design details, materials, vegetation and the variety of spaces can be modified to suit service user population demographics and site-specific needs.

## 1. Introduction

Courtyards and outdoor spaces are an essential element of acute mental health facilities, as many mental health service users are unable to leave for prolonged periods of time [1,2,3]. In some jurisdictions, such as New Zealand, the provision of outdoor space is even a legal requirement. However, the importance of quality and fit-for-purpose outdoor areas and courtyards for mental health service user health and recovery is not evident in their design. Much research has identified the link between natural light and outdoor spaces as therapeutic and rehabilitative [4,5,6,7,8]. However, although access to outdoor spaces should be established as an imperative for acute mental health facilities, many new wards are still built without direct access to external spaces, even in countries such as the UK or the US [2]. This is in spite of the many studies that have shown that access to outdoor spaces is of considerable benefit to people who are mentally unwell and can contribute directly to shorter stays [3,9,10,11]. Many established mental health units have limited courtyards spaces, often dilapidated and poorly maintained, with few natural elements or planting.

In 2021, Jenkin et al. [12] questioned the purpose of the acute mental health facility and found that whereas much was contested in terms of its function, the fundamental underpinning philosophy of care was a recovery model. The elements of recovery were described by Bird et al. [13] in the CHIME framework as connection, hope, identity, meaningfulness and empowerment. These elements of recovery can be successfully translated into design principles for acute mental health facilities themselves by adding a sixth component: safety and security, i.e., CHIMES [12]. With this modification, the framework for the establishment of a model of care has the potential to inform architectural design briefs for acute mental health facilities. However, although the recovery-oriented CHIME(S) framework (Table 1) offers a series of qualities beneficial for recovery, as it focuses on models of care, it does not provide sufficient design guidance necessary for application to courtyard and outdoor space design.

Similar to the overall acute mental health facility in New Zealand, where the purpose has been questioned [12], the role the courtyard is also questioned in terms of its contribution to recovery. Activities ranging from physical exercise and socialisation through to relaxation all feature in the literature [14,15,16]. Similarly, means of addressing contentious issues, such as the need for a place to go outside and engage with nature or the strategies for addressing for smoking, are only recently being mentioned in the literature [9]. Investigation shows that there is limited knowledge of the spatial requirements for therapeutic outdoor spaces and, in particular, those spaces necessary for individuals in confinement. Arguably, these oversights, combined with the limited amount of resources or money to accommodate service user needs, have resulted in their neglect. The aim of this research is to explore those elements of courtyard design that can contribute to the objective of supporting the recovery of the service user and to start the conversations about how we should design therapeutic outdoor spaces for those residing in acute mental health facilities.

## 2. Materials and Methods

This research adopts a ‘triangulation’ approach, interweaving multiple methodological processes, [17,18,19] including literature review, observation and interviews with service users on acute mental health units and exemplar case study analysis. The process ensures that interview material is checked against observations and the literature, and design options can be considered through comparison with case studies. The CHIMES model of recovery is used for thematic analysis.

A comprehensive search of the literature was performed using Google Scholar. In order to identify the relevant literature, alternate terms such as “outdoor spaces”, “therapeutic landscape”, “courtyard”, “outdoor environment”, “rehabilitation”, “recovery”, “therapeutic environments”, “landscape architecture”, ‘wellbeing’, ‘outdoor spaces’ and “acute mental health” were used with Boolean operators (e.g., AND, OR). Data sources generated a total of 2975 results from all the databases.

After removing duplicates (52), the remaining articles were screened using the inclusion and exclusion criteria shown in Table 2. These conditions were developed and applied to appropriately scope this review article in line with predetermined objectives.

In this in-depth review, we identified the current literature and exemplar case studies that were deemed relevant to the following research question: How can the design of outdoor/courtyard spaces of an acute mental health ward (AMHW) unit aid in the therapeutic and rehabilitative recovery of the health and wellbeing of service users and staff in these facilities? Precedent/case studies were acquired through the review of the literature and are referenced as such. The search uncovered six international exemplary courtyards/outdoor spaces, ranging from specific acute mental health therapeutic gardens to outdoor therapeutic spaces but all with a foundation in health care and/or therapeutic recovery.

As part of a larger study on the architectural design and social milieu of adult acute mental health wards in New Zealand, staff, service user and family perspectives were collected on use and experience of facility courtyards. Prior to funding, consultation was undertaken with Otago Ngāi Tahu Research Consultation Committee as per University of Otago requirements for research involving Māori. Ethics approval was received from the Central Health and Disability Ethics Committee in 2017 (17/CEN/94). The study protocol is available at http://www.anzctr.org.au/actrn12617001469303.aspx (accessed on 9 September 2022).

The interviews were conducted in four New Zealand-based facilities. These were selected to cover a range of current facilities with respect to age, size and location to better understand the framework for designing outdoor spaces in mental health facilities. The collection of interview data for this project required multiple site visits during the period of 2017–2019. As per our ethics protocol, we were provided with a list of service users on the ward who were competent to consent, well enough for the interview and potentially interested in participating. These individuals were then invited to participate in the research, and those who provided written consent were interviewed.

As part of a structured interview schedule covering a wide range of topics, service user participants were asked about their use of the courtyard and their views on its function, as well as any issues pertaining to its use. All interviews were conducted by a social scientist and experienced qualitative researcher and interviewer. Most were undertaken face to face on the ward, with a few interviews conducted by phone. For practical and budget reasons, the number of interviews conducted on each ward was capped at a maximum of ten each for staff and service users for each of the four wards. However, considerable interest in participating resulted in a few more interviews being conducted than initially decided. Interviews lasted around 30 min for service users; the interviews were audio recorded and translated verbatim.

Transcripts of interviews with service users of four mental health facilities were studied. These were initially studied to identify the main concerns with respect the facility and courtyard design. Extracts from the transcripts were coded into themes separately by researchers using the iterative process described by Braun and Clarke [20]. All researchers then met to compare and agree on the main themes. Issues need an expansion of ideas to inform solutions, whereas positives can be celebrated and developed. Therefore, the interviews were also searched for mentions of positive elements of current facilities that could be used to inform the future of mental health ward/courtyard design.

## 3. Results

Following the CHIME(S) framework, findings were organised according to the themes of connectedness, hope and optimism, identity, meaning and purpose, empowerment and safety and security.

### 3.1. Connectedness

In most acute mental health facilities in New Zealand, the courtyard is the only type of outdoor space that service users have access to, as many are unable to leave for a significant portion of their stay. Access is usually limited to daytime hours. The courtyards of acute mental health facilities are commonly designed with an excess of hard surfaces, limited planting and cage-like security screening (Figure 1), as hard surfaces make it difficult to hide things and are easy to keep clean. This is reportedly for the safety of staff and the public, as well as for ease of maintenance [21]. It is not uncommon to find courtyards used for illicit activities, such as smoking, which is largely prohibited in healthcare environments. Few, if any, courtyards of acute mental health facilities in New Zealand are used for formal therapeutic activities.

Many studies report that nature-based settings foster connection through social interaction [22,23] and that recreational amenities and outdoor spaces often increase the sense of community and interaction [24,25]. Furthermore, research shows that people who are connected to nature report higher levels of environmental consciousness, happiness, social wellbeing, acceptance and sense of self [26,27,28,29]. With respect to courtyard seating, sociopetal seating arrangements (people facing people) and mixed arrangements have been found to significantly contribute to more personal interactions when compared with unstructured or sociofugal arrangements [30]. More specific to mental health, research finds that people who are experiencing crisis are less affected by the crisis when in frequent contact with nature compared with those with less contact with nature [31].

For heightened quality of life while in care, service users require the friendship of other service users, they need to be able to host visitors, and they need to productively live with others. Whereas most acute mental health facilities are designed for short stays of 2–3 weeks, the reality is that some are residents for months, and a number of service users are regulars, returning as need arises. For durations of more than a couple of weeks, larger courtyards or unsupervised outdoor access is essential. This can be addressed through attention to the size and variety of courtyards and experiences. Facilitating small groups through seating and landscaping can foster more intimate relationships both between staff and service users, as well as among service users. To ensure connectedness to both self and nature, courtyard space must be near common areas or adjacent to them, and service users who are too unwell to use these spaces should be placed in rooms overlooking the courtyard so they can appreciate and benefit from the natural stimulants of the space [32].

In the Aotearoa New Zealand context, participants reported on the importance of the courtyard in terms of making connections with others but also highlighting how poor design and use negatively impacted their connections.


*A008 “So the patients were all sort of counselling each other really... yeah a lot of talk goes on out in the courtyards. I try and look after all the... I’m usually one of the older people there, so I try and look after the young ones that come in.”*



*A003 “I talk to other people usually. But quite often I just sit alone, just for the peace and quiet, to be out in the fresh air. And yet still potentially in contact with people even though they don’t make an [inaudible] to me and I don’t make an [inaudible] to them. It’s still nice to have people around even if you’re not talking to them. So you get your quiet space but with people around. So you can talk to them if you like.”*



*C005 “That was one of those was quiet. That one of these yeah. Pretty good. There may be there’s another one out there the tranquil gardens. Yeah, that one was mainly used for smoking people, the girls sometimes went out there having a girlie chat Let’s make sure that sometimes those young male Māori guys have had their friends around and yeah get out the here that was well which was like problems out there seemed to be fine.”*


A good example of design that promotes connectedness is Alnarp Rehabilitation Garden in Sweden (Figure 2). The 2 ha garden provides the opportunity for connectedness through the organisation of intersecting pathways and the creation of small ‘rooms’ for contemplation or for social engagement.

Designed especially for those diagnosed with depression, the Alnarp garden allows for an immersive experience in the carefully designed landscape setting, focusing on a sensorial encounter with textures, plants, smells, sounds of nature and variations in light quality from brightness to shadow. Within the larger garden setting, several smaller ‘courtyards’ provide more intimate opportunities for contrast and variation, allowing for rest and contemplation, as well as social interaction and solitude. A range of therapeutic activities are undertaken here, both formal and informal, supervised and unsupervised, ranging from viewing others from afar to working alongside others. Spaces have been developed to allow users to take refuge in but also to look out from so that they are able to survey the landscape and gain control over their interactions with others [6]. Activities vary in intensity to stimulate curiosity and foster meaningful engagement across a range of service users.

Appropriately designed courtyards can aid in the provision of care through their service to the wider goal of recovery. The design and layout of these outdoor areas in mental health facilities can influence social interaction and can enhance the sense of togetherness and community [33,34], thereby addressing the need for connectedness.

### 3.2. Hope and Optimism

In contemporary mental health practice, recovery orientation is evolving as the new service paradigm, for which hope is central [35]. A sense of hope and optimism is a future-oriented expectation that attaining personally valued goals can give meaning to life. Potentially informed by negative experiences, such as mental disorder, feelings of hope and optimism are considered possible but depend on personal activity, as well as external factors [36,37]. Feelings of hope and optimism can be a trigger for recovery but can also be a maintaining factor, as they help people to both find the courage to start the process of recovery and maintain motivation to keep working on recovery despite obstacles [38]. Hope is consistently identified by both service users and therapists in various settings as a key factor in psychotherapy [36] and is deemed essential for resilience [39].

Creating an environment that fosters hope and optimism is therefore deemed to be essential to aid in the motivation for change and the belief in the possibility of recovery. In terms of courtyard design, views of nature and gardens were also found to aid in reducing stress and pain [40]. Service users with access to windows experience fewer sleep and visual disturbances and suffer less from hallucinations and delusions [41]. This is important, as recovery entails having dreams, aspirations and positive thoughts for the future, many of which are formulated during the quiet times prior to and following sleep.

A lack of activity engagement often results from medications prescribed for some mental illnesses that slow people down and make them gain weight, leading to a lack of sleep [42]. Sleep deprivation is common in mental health facilities, as little to no physical effort is expended during the day, and the facilities are often noisy [2]. A further benefit of designing opportunities for engagement in activities is that such interests are infectious (so there is a social benefit), and they are ongoing. Vulnerable people can take these ideas and interests with them wherever they go, and these skills will prove protective against unwanted automaticity, paranoia and other symptoms.

Access to natural light is widely reported as beneficial [27,43,44]. Healthcare consultants agree that natural daylight improves sleep habits and sleep patterns for mental health service users on psychiatric medication. Similarly, warmth and sunshine in a facility balanced and harmonised with both calming and stimulating spaces can help reduce distress and relieve pain. Hope and optimism can be fostered by an initial multisensory experience with variation in colour, shape, smell and texture. Plant selection with consideration of pleasant smells and tactile textures both underfoot and at hand level has been found to have a relaxing effect on muscles, improve concentration and enhance the production of endorphins [45,46], whereas unpleasant smells can be associated with anxiety, stress and fear [47]. Smell can also affect people’s mood, memory and behaviour [48]. Protection from the elements, shade and shelter, along with space allocated to open views, means that courtyard spaces can be used year-round [32].

In Aotearoa New Zealand, services users describe their only outdoor experience with mixed feelings, welcoming vegetation, seating and the sense of a garden but feeling frustration with the design, the materials used and the dominance of smoking. The design and age of the furniture was also considered to be a problem:


*A004 “Oh God, even if they brought some false grass in, you know, that grass, that stuff? Because it’s like concrete there and then when it’s hot there’s no shade. …there is one shade cloth, which is a waste of time having it there… there’s no shade”.*



*A009 “Yeah, the picnic tables. You sit on one side and the other side goes up and your drink goes flying. It’s the same ones since I was 19. They’re still the same fricking chairs”*



*B003 “…used to be a gazebo…but someone broke it.”*


An example of design that promotes hope and optimism is Dannerhuset Healing Garden in Denmark (Figure 3). The 0.1 ha garden incorporates nature-based healing strategies to provide respite to female and child survivors of domestic violence.

This crisis shelter garden is designed to provide opportunities for privacy and refuge while remaining connected to its surroundings. Daily activities and therapeutic sessions in the garden aim to reduce stress and anxiety and simultaneously increase bodily awareness, aid in the development of self-worth and promote belief in the possibility of recovery. Formal and informal design strategies are employed through the use of softened landscape elements, enclosure, exposure opportunities and variations in scale to promote positive thinking and foster dreams and aspirations. The tactile and olfactory properties of plants provide an opportunity for experiential engagement with self and with others. The relationship between vegetation and other soft or hard components of the landscape creates spatial variability and sensory engagement and improves ecological health.

Although current models of care in many acute mental health facilities promote a low-stimulus courtyard environment, evidence shows that outdoor spaces with more features and stimulants provide the chance to create a greater sense of home, which is important for hope and optimism. Variety in design can create multiple possibilities of contact and therapeutic scenarios, aiding in recovery.

### 3.3. Identity

Cultural orientation and social preference can dictate specific requirements for outdoor spaces, generating a spatial need to accommodate diverse types of activity. The space must accommodate the needs of different cultures and ethnicities, gender orientations, social habits and different mental illnesses. These groups all have particular requirements that need to be satisfied by a particular space; therefore, there must be an abundance of both opportunity and activity for all groups of people and individuals alike.

Dijkstra et al. [41] discuss the benefits of physical environmental stimuli in healthcare settings with respect to the health and wellbeing of service users. Native plants can also feel familiar to the service user and create a sense of ‘home’ while creating natural stimulants that aid in the recovery of service users’ health and wellbeing. Creating a home-like appeal within an outdoor space can be difficult, depending on the identity of its users, as many individual differences must be considered. For example, people from collective cultures may prefer socially oriented areas, whereas those from more individualist cultures may seek areas of aloneness, and personal distance preferences can also vary by culture. At the level of the individual, many preference variations result from personality types, values, age and gender, as well as mental health presentation and symptoms. Attention to identity in the outdoor environment therefore requires the use of multiple environmental stimuli to support the wellbeing of service users.

To address the unique needs of individuals, dualities of diversion and contemplation demand dynamic interactive spaces that allow for escape and stimulation, as well as the opportunity to sit quietly and alone, meditate, gaze on nature and foster open-ended imagery that is rich for the senses. Social interaction can be fostered through the design of a multiuse space that has a broad-based appeal but can be balanced with nurturing spaces that are family-centred with culturally appropriate imagery. Spatial order and orientation can aid in establishing trust and confidence at the beginning of a journey, as well as cultural sensitivity and invitation to participate, hope, renew and feel optimism.

In the Aotearoa New Zealand context, the imperatives of biculturalism demand equitable attention to Māori culture. Unlike Western approaches to medicine, Māori take a holistic view, with a focus on health preservation and the improvement of the body to resist disease, which emphasises a sustainable connection between nature and humans [49,50]. Whereas theoretical frameworks for healing landscapes are relatively well-established in Western countries, there is a significant lack of published research drawing on Indigenous knowledge for the practical design of healing gardens. Incorporating Indigenous values and approaches can facilitate a restorative and therapeutic landscape and offer new opportunities for living with nature and supporting health and wellbeing. Positively working with local *iwi* (Māori tribe) and Indigenous groups and embedding their values into design can offer outstanding opportunities that benefit all.


*D007 “He said the space was really lovely, and um, but the thing that was lacking as he goes, ‘there’s ten of us—males, in this ward…those colouring in and mindfulness stuff, that’s what my baby does. And I’m forty. And I don’t want to be colouring in to be honest, is there any activities ‘cause the few of us and he looked at, you know at the others, [and said] “A few of us we’ve done time in prison. Yeah. We don’t want to be doing colouring in.”*



*A009 “Yeah. I reckon it needs a bit more nature out there. There’s nothing out there. It’s just plain. It looks like a bit of a schoolyard. You’ve got two little veggie boxes, but that’s not enough”.*



*B003 “It’s still nice to have people around even if you’re not talking to them. So you get your quiet space but with people around. So you can talk to them if you like.”*



*D004 “Like play tennis or like badminton, things like squash, like get them fit, like have a little gym…We just need a rugby ball so we can um, play rugby or a tennis ball and all that. Even like a hoop would help.”*


An outdoor example that addresses designing for identity is the Kopupaka Wetland Reserve in Aotearoa New Zealand (Figure 4).

Fundamentally designed to enable stormwater management, the 22 ha park focuses on the ecological restoration of a stream by bringing forth the site’s associated Indigenous values. These are intertwined with cultural and recreational aims to foster healthy landscapes, which in turn contribute to healthy individuals, an essential concept for Māori. Meeting places, active play areas, open fields, a botanical weaving garden and horticultural planting are balanced with places for individual contemplation and retreat, providing both refuge and outlook. The senses are engaged through views over wetlands; active birdlife; and accompanying sights, sounds and smells, whereas the body is engaged through sitting or walking in hot sunshine or cooling shadow. The park celebrates the inseparable bond of water and earth, vital to sustaining and balancing the natural environment while permitting Māori and non-Māori users to regain trust and rebuild and redefine a positive sense of identity. Here, the landscape affords connection with others, such as extended family, as well as a shared understanding of nature, creating an inclusive environment that emphasises development and the maintenance of relationships through respect and trust.

### 3.4. Meaning and Purpose

A growing evidence base suggests that having a strong sense of meaning and purpose in daily activities can imbue one’s life with a sense of worth and foster health and wellbeing [51]. Basic human needs are fulfilled through choice, control and belonging [52]. This is acknowledged by the WHO (World Health Organization), which finds that performing meaningful activities is pivotal in enabling participation in major life areas. Whereas the need for engagement with activities deemed meaningful and purposeful is considered crucial to the recovery process, evidence is scarce with respect to how to support persons in re-engaging and sustaining meaningful activity engagement [42].

Outdoor space can facilitate a wide range of therapies and uses for different service user groups, as well as therapies that are beneficial to both individuals and differing groups at the same time [32,53]. The design of outdoor space is also important to encourage participation in physical activity [54]. For example, Joseph et al. found that an environment with a clear destination point and a well-structured pathway network designed for social interaction had a positive effect. Pathways designed for recreational walking need to be longer, less fragmented, without steps and with attractive views to result in greater use [54]. Exposure to nature is one key aspect of participation in outdoor activities that provides a range of health benefits, including the perception of barriers and relief from stress [3,55,56,57,58,59,60,61], improved physical health [62,63] and mental wellbeing [64,65]. Time outdoors and time in nature plays a critical role in bolstering resilience to stressors, underscoring the need to facilitate outdoor recreation opportunities, especially during times of crisis [66]. Whereas even simple exposure to outdoor green space has an effect on a service user’s psyche, for a courtyard to offer meaning and purpose, it must first be sufficiently large to offer a range of experiences. Solely walking the perimeter of a courtyard or grounds can become repetitive and boring.

The activity in a courtyard does not have to be purely physical activity. Meaning and purpose can be aided with facilities such as small outdoor ‘rooms’ to allow the shared exploration of the meaning of the mental illness experience with other service users, with spaces designed for contemplation, confession, spirituality and prayer, as well as spaces for staying fit and able, contributing to the community or a special interest group. A cohesive group of microspaces designed for each courtyard can extend the range of activities carried out. Socialising, sitting, thinking, sleeping, etc., all need to be considered when it comes to designing a courtyard for a mental health facility.

Comments by service users support the literature:


*A009 “Yeah it’s… I don’t know, it seems to be the only place everyone congregates and gets together and whatnot, but it’s dirty it’s gross. It’s...the chairs are shit, there’s not enough chairs actually, for everyone. And… I mean the basketball hoop’s good, you know, but maybe I don’t know they could put something else out there. I don’t really know... I’m not a big sports person”.*



*A007 “[Go in the courtyard] Yeah because there’s nothing else to do in here, it’s so boring, it’s depressing, it gets you down.”*



*A004 “I spend the time basically walking around and try to keep fit, in the courtyard even though the courtyard’s all concrete it’s somewhere. Somewhere to just walk around and try to keep fit. It can be quite boring you know.”*


Nacadia Healing Garden (Figure 5) in Denmark addresses the concepts of meaning and purpose through the incorporation of a nature-based healing strategy [67,68].

This 1 ha garden is designed to understand how and why nature has a positive influence on people with stress-related disorders from a preventive and treatment perspective. The restorative naturalised and structured areas for activity-based horticultural therapies are salutogenic and offer supportive functions, with social spaces designed for vulnerable people. The design varies from a large scale (e.g., forest) to a more human scale (e.g., user) to provide a refuge, a serene environment with richness in species and settings for different levels of communication and participation. The garden is designed as a sum of various ‘courtyards’, i.e., open, semi-open, closed and enclosed spaces that provide different involvement levels. The first level of engagement is about observing nature and being alone. The second is to introduce activities, such as gardening and harvesting food. The final level of engagement seeks to forge social interaction and increase the levels of participation and involvement. Due to the sensitivity of users, smaller spaces for retreat incorporate experiential qualities, such as wilderness, peace and safety, transitioning to open social spaces as mental strength increases.

The sounds of nature, birds and water, as well as the mirroring in the water, keeps the user in the present, focusing on positive moments that allow them to get away from stressful thoughts of the past or the future. The glasshouse acts as a hub for rehabilitation. Gardening activities, gathering places, closed rooms, water elements, a dining table, a fireplace and a hammock all contribute to different levels of participation, ranging from being alone to being involved in more social activities. Vegetable gardening is also available next to the outdoor kitchen. This kind of activity is social, meaningful and creative, which makes the user feel rewarded. The overall experience fosters spiritual connection, offers meaningful activities and provides a more vital purpose for a better quality of life.

### 3.5. Empowerment

Research reveals that dignity, confidence and a sense of control and autonomy may be subtly reinforced when service users make their own decisions about how they will spend their time outside [69]. In terms of design, this means that outdoor spaces must offer choices for service users, staff and visitors alike. There could be courtyards where families can visit and others for service users only. By offering choice, there is little limitation as to the activities that can be carried out within the outdoor space. Attention to outdoor microspaces with specific design objectives in mind is essential to the crucial programming of these spaces if they are to target a variety different service user needs. For these reasons, courtyards and other outdoor areas should be designed for multiple activities with the ability to adapt to service user presentations and needs.

There is a difference between visual and physical access. Service users need to be able to view and survey the courtyard to feel empowered to be outdoors. They need to be able to easily find their way to the outdoor space, enter the physical accessway (as some are locked) and inhabit the courtyard. If there is only a view to the outdoors but physical access is denied, this can be detrimental to the health of users, representing a critical issue for 95% of service users [70]. Service users need to be able to independently explore the area to truly benefit from their interaction with nature in a manner that will aid in their recovery. If a facility is to invest in high-quality landscaping, there is really no point if it is not accessible. Lack of outdoor access can feel like a form of psychological torture when outdoor space is in view but out of reach [71].

Whereas research has also demonstrated that access to therapeutic outdoor areas and even viewing them can speed up the healing process by improving the effectiveness of the treatment that these facilities provide [63,65], Ulrich et al. [72] found that physical access to nature is more effective than simply views of it and that even limited periods of physical access reduced stress and anger for both staff and for service users. Inaccessible or locked courtyards were found to decrease feelings of control and instead lead to an increase in stress levels. Confronted with a multitude of options, service users must control their own choices, whereby self-esteem is subconsciously reinforced [69]. If service users make their own decisions about how they access and use such spaces, levels of stress and anxiety can be tempered so that the courtyard is actually beneficial to their health and wellbeing [69].


*B006 “Well I was on edge all the time which is why I’m in here but this has made it completely worse. Breeze, grass, outdoor area, somewhere I can actually just walk around looking at birds, or listening to birds. There’s no access to real outside. That’s a courtyard area that’s all completely; it’s in the central, centre. So you can’t really, it’s just building all around.”*



*A010 “Yeah it’s actually quite a nice courtyard, it’s like being able to have a few plants out there, which is nice. It’s got very dirty though, because the story was people were occasionally feeding the pigeons. The pigeons were pooping, and then in theory possibly it was causing fly problems as well. And it was pretty dirty with cigarette butts and stuff. So I got one of the staff members and then said can I have a chance to clean that part of the courtyard. So he let me use this fire extinguisher and I gave it a complete blast to try and clean it.”*



*A001 “Yeah. I mean the plants were pretty much dead. And a couple of people took it into themselves to wash the court yard with a hose and scrub brush. So they did a really good job.”*


Maggie’s Care Centre in Manchester (Figure 6) is designed around principles of rehabilitation and empowerment for those recovering from cancer.

The 1.9 ha building complex and garden provide a welcoming ‘home away from home’ through communal areas within which visitors can find private places for emotional retreat. Both the internal and external spaces are designed to set up very ordinary encounters and facilitate acts of kindness between healthcare professionals, volunteers and visitors that nurture personal recovery. The open core of the Centre is designed to foster human connection both communally and individually. Private spaces are located in the periphery, timber is used to create a feeling of warmth, the extensive use of glass facilitates permeability and exposure to the outdoors and the scale of the building is human. Volunteers and visitors can have a physical experience through sitting, walking or gardening. Both the internal and external spaces provide an empowering experience to service users, as they can exercise choice and control at a time at which these options are not so readily available.

### 3.6. Safety and Security

Among the themes of recovery, safety and security are perhaps the most challenging for designers due to the focus on controlling as opposed to enabling. Advances in technology provide new freedoms in design; unobtrusive CCTV surveillance can work to dispel perceived risk in contexts where privacy is perceived to equate with danger, with concerns for blind spots, danger spots and unimpeded lines of sight, as well as the increased watchfulness over service users’ movements through less controlled space [21]. One potential risk associated with this approach is that recording untoward events rather than preventing them may become the security engagement, with the recording becoming an end in itself. The challenge is to design spaces of care that offer reasonable levels of technical safety without defaulting to locked doors. Relational security and environmental visibility are what is needed, in addition to the flexibility to accommodate changes in practice. A possible future trend might be toward practice guidelines that allow for ‘therapeutic risk taking’ negotiated in a way that empathises with service users’ experiences and feelings.

The size of the courtyard can be important to its functionality, as low social density is key for stress reduction and, by extension, can reduce aggressive behaviour. People with a history of aggressive behaviour need significantly more personal space than others, and the experience of perceived violations of personal space are amplified in small spaces. Service users reported on the negative aspects of safety and security, which, rather than making them feel safe, made them feel alienated. Feelings of security can also be enhanced through access to natural stimulants, such as daylight and activity with nature. In a material context, many naturally forming resources can be used to improve the health and wellbeing of service users. Timber and stone contribute more to mental health than large concrete walls boxing in the courtyard space [32]. Other materials include vegetation and water features, each associated with requirements and limitations. These materials are calming and stress-free, whereas the social stigma around large concrete walls topped with barbed wire makes for a harsh and cold environment. Prison-like design insinuates that service users are locked-up and forgotten about, and with no natural stimulants to aid in recovery, this detrimental design typology does not support the mental health of service users within a mental health ward.

In addition to more specific anti-ligature design strategies, key elements include natural surveillance and natural control access with direct sight lines; attention to spatial ordering, territorial reinforcement and clear spatial delineation that enables users to develop a sense of proprietorship; and good maintenance that demonstrates that a space is not neglected [73,74].


*B004 “[on a courtyard in West Wing] it’s got a beautiful garden to look out on. Quite a substantial garden, with trees. And I think they used to have pagodas in the courtyard. And then people were trying to hang themselves, and they had to go. So they took all those away. This one’s only got a basketball court, which is enough to drive you bonkers. Like a hospital for mentally ill people, and someone’s out there bashing a basketball. I mean that’s the last thing you need”.*



*A002 “There’s nowhere else to sit except out in the courtyard, or stand in the courtyard basically because the tables are covered in cigarette butts and disgusting things, it’s so unsafe. So there’s courtyard and basically all the wards and all the offices and everything goes around this courtyard. That courtyard could be much better utilized. I could see probably an extra six rooms are there, you know if you got rid of the courtyard and made just a little smoking area, you know just a little smoking area away from that and make the visitors areas, so you actually had your family there”.*



*B004 “Initially I thought I really liked the garden- big lovely garden with trees, and chairs under it. The worst thing about- you know, that one there- the one we’ve got here, this courtyard here is not great, but at least you’re not looking at a wire cage, like being in High Care Area, like you’re being a tiger in a cage. And the motorway’s right there, and everyone’s driving right past you. Yeah, so that was a bit awful”.*


The recently completed acute mental health unit of Belfast City Hospital (Figure 7) is designed to promote the safety and security of service users through connection to the outdoors and nature.

The 80 ensuite bedrooms, including six psychiatric intensive care units, are organised in five separated yet interconnected buildings around a cloistered central and circular courtyard and communal and administrative areas. The idea of the courtyard was to ensure that the surrounding environment would enhance the service user experience and their mental health. A range of materials, colours, lighting, furniture and fixtures was carefully considered in the design. Various activities became the main focal point through gardens, planting and the placement of seating and tables.

The Acute Mental Health ward in Belfast City Hospital illustrates how best to immerse service users within nature to create a better healing space. It is made up of many different gardens dedicated to different people but all with the benefit of healing. Walking through the building, a sense of calmness is noticed. The idea behind the design was to give suitable service users the opportunity for outdoor recreation and their own leisure. This nature-based design draws inspiration from its connecting gardens rather than long walkways and corridors. All aspects of service user life are centred around a courtyard so that everyone who experiences this facility has meaningful contact with natural elements to foster safety, trust and resilience.

## 4. Discussion

In this study, we triangulated data from published sources with case study examples and the voices of staff and service users. Defining the problems from inside acute mental health facilities worked to truly gauge how service users felt about the space in which they resided. Their important perspectives, combined with findings from the literature and expressed in successful case studies, can help to holistically inform designers. Along with empirical research on the value of courtyard spaces, there is now sufficient evidence to inform decisions about design specifics, such as which elements, configurations and programming in outdoor spaces can elicit the best possible outcomes. Unfortunately, this information does not often make it from research into practice [75].

To address this situation, the CHIMES framework, which was developed as a recovery model for care, was tested as a method of analysis and a means of grouping ideas in order to develop a foundation for design that goes beyond prescription and addresses the essential qualities of courtyard space that are required for recovery. This finding aligns with that of leading theories and research from the US and Scandinavia (Table 3).

Research on outdoor environments at crisis shelters revealed similar themes. Safety and security concerns were addressed through the demarcation of spaces and oversight, as well as enhanced lighting. Sensory aspects of the garden, such as sounds, colours and scents, were welcomed, as was seasonal change [76]. This research also found that meaningful activities need to take into consideration the duration of the service users’ stay. The length of stay in the crisis shelter was typically short, and many service users were only there for a week or even less, so did not participate in activities associated with longer durations, such as gardening.

The CHIMES themes were particularly useful in grouping and assessing the necessary qualities of the courtyard spaces. Connectedness extends to several levels of connection: the fostering of relationship development and support from others, the facilitation of peer support and support groups and connection to the wider community. Creating an environment that fosters hope and optimism is deemed to be essential to aid in the motivation for change and the belief in the possibility of recovery. The establishment of identity and the redefinition of a positive sense of identity, together with the journey of rebuilding it, are also part of the recovery process. Our research is situated in Aotearoa New Zealand, a bicultural country where both Western and Indigenous ideologies are acknowledged to develop culturally appropriate and theoretically sound outcomes in the design of the built environment. Identity can be fostered by providing options giving autonomy and choice. Fostering meaning and purpose can mean continuing on with the familiar but also trying out new activities. In our research, limited governmental investment was deemed responsible for both a lack of suitable facilities for meaningful activities and a lack of supervision. Spaces designed for activity were often locked or unused due to staff shortages [16]. When there is a varying typology in a spatial design, there is no limit to the number of interactions a service user can have within said space. This prevents feelings of repetition and can move the user to interact differently with the courtyard itself. The provision of a suitable courtyard can empower individuals by allowing them to focus on their strengths, facilitating personal responsibility and control over life by allowing service users to maintain the outdoor space. The landscape architectural implications of the CHIMES model for courtyard spaces are summarized in Table 4.

Whereas the CHIMES framework is beneficial as a reminder for all of the qualities that are necessary for a recovery environment, it is not a precise science, and there is overlap between the themes. This is particularly prevalent when considering architectural materiality. In terms of design, the materials of a space inform how we perceive the type of space it aims to be, for example, finding a happy medium between greenery and concrete courts. Here, the main issue is the balance with respect to variety of activity; if there are not enough options for service users, then these spaces will not be used. Excessive use of tectonic materials, such as concrete and steel, can have negative health impacts on clinical disorders such as depression and anxiety. There is also a social stigma around these materials, as they are likened to prison architecture. The aesthetic style of facilities is important, as it reflects how society feels about such places.

Whereas there is a growing body of research on the efficacy of healing garden benefits to service users, visitors and staff, the scientific background of natural stimulants and phytoncides is an area of future interest. The neurobiology of plants and other natural phenomena and how they affect the brain could inform the best use and type of both plants and vegetation within a courtyard.

## 5. Conclusions

The success of restorative courtyards is critical for many reasons. First, a well-designed, well-maintained courtyard promotes the best possible health outcomes, both physical and mental, which has been widely documented internationally across a range of healthcare facilities. Acute mental health service users are among the most vulnerable members of the population, and if recovery is the objective, access to the outdoors and nature is paramount. One of the most significant contributing factors to a service user’s mental health is the environment by which they find themselves surrounded. Nature and natural materials affect cognitive function, emotional wellbeing and other aspects of mental health. The research reveals that a good, varying, healing environment is far more beneficial to a service user’s mental state than some of the current facility designs. The materials used in most existing facilities are large, concrete walls topped with wire and with cage-like enclosures. This material selection is detrimental to the health and wellbeing of its occupants, service users and staff alike. Courtyards are meant to aid in mental health recovery rather than contribute to mental distress.

Second, a courtyard can offer a positive image of the facility for all stakeholders. The social stigma of ‘prison-like’ designs can be minimized by the selection of natural materials that aid in therapeutic recovery, such as timber, stone, vegetation, water features and artwork. All these features can be designed in a cohesive web of micro- and macrospaces to provide more variety at an even larger scale.

Third, every successful restorative courtyard is a powerful testimony to the restorative abilities of gardens and access to nature as a whole. With respect to the typology of a courtyard, the main idea to keep in mind is the aim of the space. Research clearly suggests that a lack of variety within courtyard design makes these spaces unusable and unwanted. The idea of variety recurs in all aspects of the research, especially when it comes to the way in which courtyards in mental health facilities are designed. There can be no limits to the number of interactions service users can have in such spaces due to a multitude of uses. The variety that these spaces can accommodate both multiple interaction possibilities and different types of service users. Groups such as those from different ethnicities, different genders and with different mental illness presentations all have specific needs. Therefore, there is a requirement for an abundance of spatial design. Finally, it is important to regularly audit courtyards in acute mental health facilities to understand why some succeed and some fail. Unless this happens in a consistent way, designers cannot learn from past mistakes.

## Figures and Tables

**Figure 1 ijerph-19-11414-f001:**
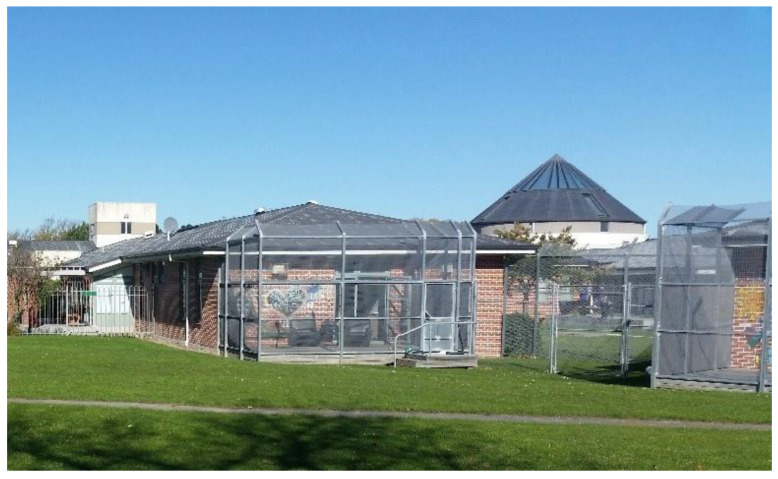
Cage-like screening in an acute mental health ward.

**Figure 2 ijerph-19-11414-f002:**
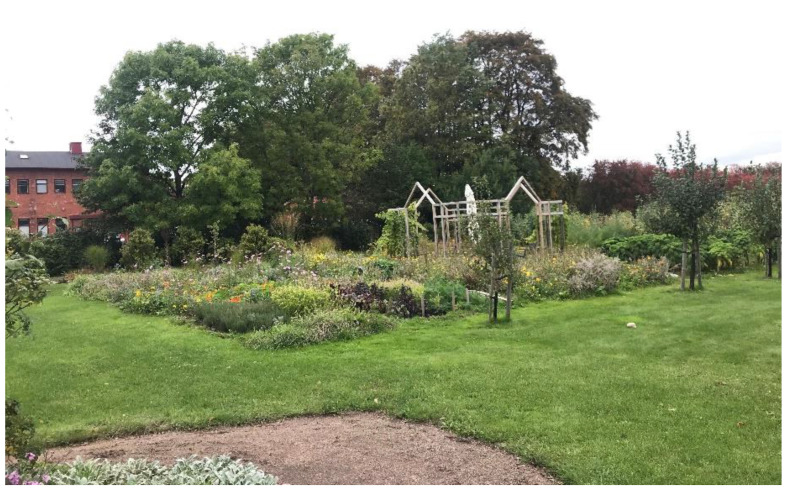
Alnarp Rehabilitation Garden, Sweden.

**Figure 3 ijerph-19-11414-f003:**
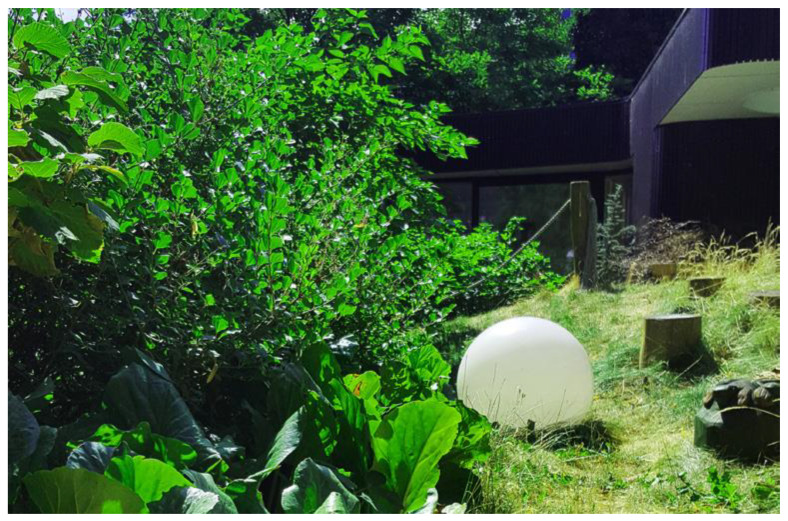
Dannerhuset Healing Garden, Denmark.

**Figure 4 ijerph-19-11414-f004:**
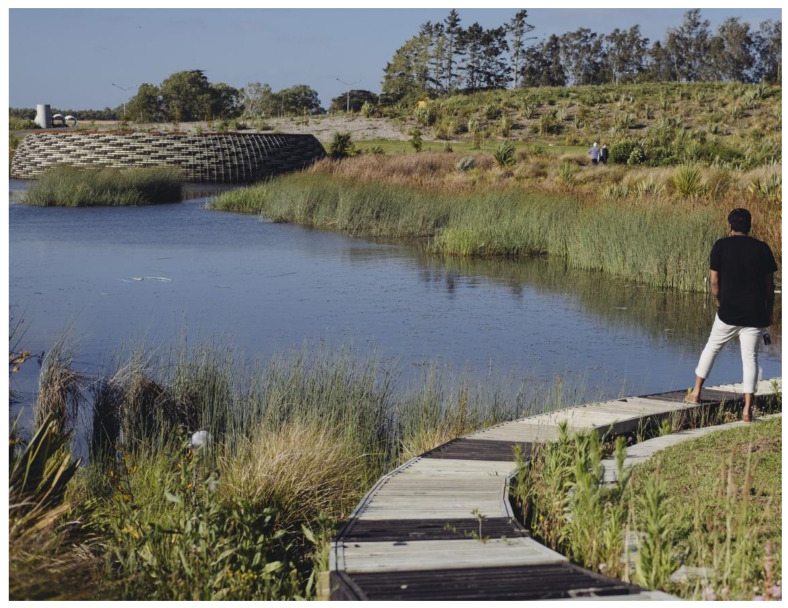
Kopupaka Reserve, Aotearoa New Zealand.

**Figure 5 ijerph-19-11414-f005:**
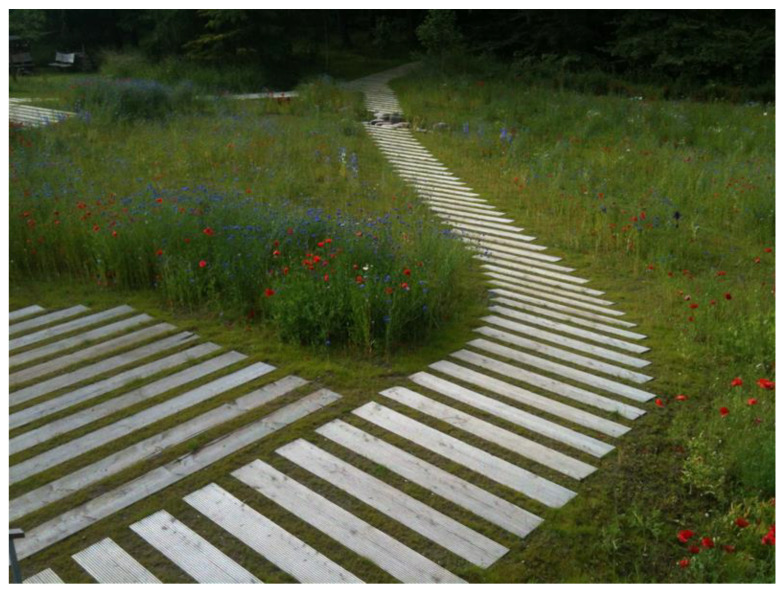
Nacadia Healing Garden, Denmark.

**Figure 6 ijerph-19-11414-f006:**
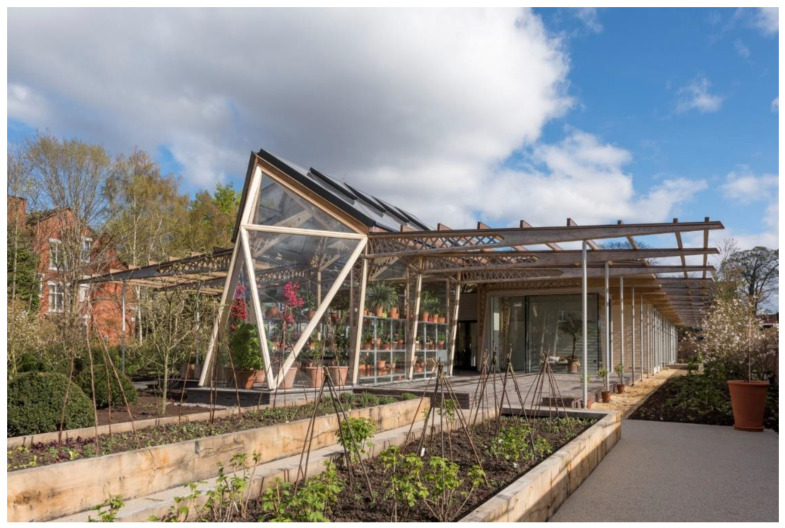
Maggie’s Cancer Care Centre in Manchester, England.

**Figure 7 ijerph-19-11414-f007:**
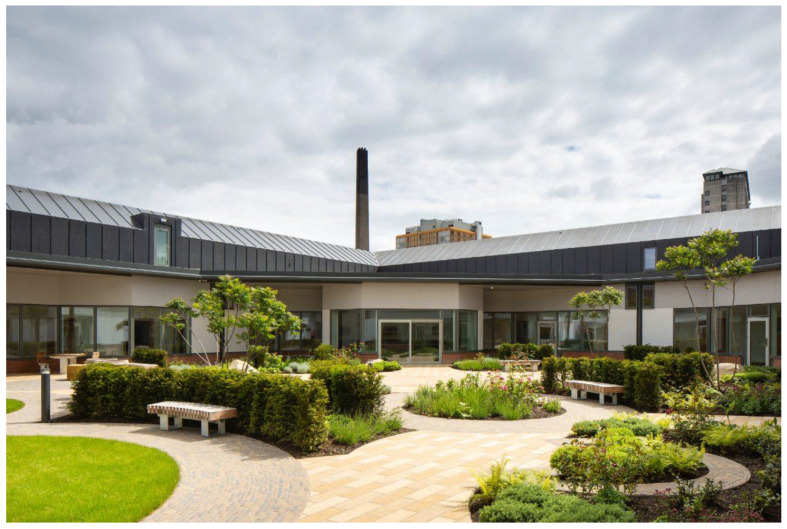
Acute Mental Health Ward in Belfast City Hospital, Northern Ireland.

**Table 1 ijerph-19-11414-t001:** CHIMES model of recovery and exemplars.

Theme	Service Implications
Connectedness	Providing both company and privacy Fostering Peer support and peer groups Relationships Support from others Being part of the community
Hope and Optimism	Providing respite from symptoms Belief in possibility of recovery Fostering Positive thinking Dreams and aspirations Hope inspiring relationships
Identity	Overcoming stigma Rebuilding or redefining a positive sense of identity
Meaning and Purpose	Providing quality of life Fostering Spirituality Meaningful activity Rebuilding of life
Empowerment	Providing choice in personal recovery Fostering Control over life (transparent rules on the ward) Focus on strengths and interests
Safety and Security	Providing safety and security

**Table 2 ijerph-19-11414-t002:** Inclusion and exclusion criteria.

Inclusion Criteria	Exclusion Criteria
Peer-reviewed articles (2010–2022) Empirical research English-language literature Outdoor built environment Acute mental health	Non-peer-reviewed articles Conference proceedings Narrative reviews, lecture notes and studies published in theses Non-health-related studies

**Table 3 ijerph-19-11414-t003:** A comparison of leading US and Scandinavian models and frameworks with the CHIMES model (adapted from [3,75]).

CHIMES Themes	US Model	Scandinavian Model	Description
Connectedness	Nature distraction/engagement		High ratio of green landscape; rich sensory details; seasonal interest; sight and sound of water; wildlife.
Hope and Optimism	Emotional and physical comfort Adequate maintenance	A range/balance of experiences to address a range of mental health issues (e.g., walking on different ground surfaces to increase body awareness and provide exercise); accessibility for people with functional disorders.	Covered seating at garden entry; comfortable seating throughout the garden; mitigation of extreme weather; quiet location; regularly well-maintained garden.
Identity	Social connection and support	Attention to geographical and historical context and initial experience; familiar feeling over abstraction; serene and spacious with richness in species and culture; restorative ‘room’(s).	Semi-private seating clusters; close proximity to nursing units, waiting rooms and staff break rooms.
Meaningful Activity	Physical movement and exercise	Designed for horticultural therapy program; inclusion of areas that are purposefully left ‘unfinished’ to empower service users; provision for levels of involvement ranging from inward reflection to outgoing involvement.	Level, non-glare pathways with appropriate traction; destination points.
Empowerment	Sense of control Visual and physical accessibility	Ensure ‘rooms’ where sad, distressed and upset people can calm down and be restored.	Moveable seating; variety of walking loops; places to sit in sun or in shade; ensure garden is visible from well-used indoor spaces, such as lobbies and waiting rooms; doors and thresholds to garden are easily navigable.
Safety and Security	Safety, security and privacy	The creation of ‘rooms’ to clearly delineate parts of the courtyard from the surrounding spaces and increase the feeling of security.	Clear boundaries or sense of enclosure; places for people to retreat on their own or with others; adequate lighting.

**Table 4 ijerph-19-11414-t004:** List of needs that courtyard design must accommodate (adapted from [13]).

CHIMES Themes	Landscape Architecture Implications
Connectedness	Outdoor spaces for family and friends to visit Sociopetal furnishing layout Family support areas Space to contribute to the community
Hope and Optimism	Access to nature Access to natural light Mix of calming and stimulating environments Quality/supportive environments for staff to decompress Informal and home-like Easy to maintain
Identity	Options giving autonomy and choice Stimulation/solitude Family-oriented/individual-oriented social interaction Maintenance of possessions Sense of ownership
Meaningful Activity	Rooms for contemplation, spiritual connection and prayer Family support areas Space to stay fit and able Space for privacy Space to contribute to the community Space to continue meaningful activities Opportunities to try new activities Common areas where productive activities can take place, such as: Growing food; Gardening; Outdoor cooking; Undertaking community-building activities; and Playing sports
Empowerment	Selection of options to focus on individual strengths, such as: Making art and music; Learning; Hobbies; Gardening, cooking and caring for animals; and Exercising Moving around Orientation and wayfinding
Safety and Security	Unobtrusive technology Direct sight lines Clear spatial delineation and demarcation Attention to spatial ordering

## Data Availability

Data presented in this study are available upon request from the corresponding author. The data are not publicly available due to ethical and privacy restrictions.

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
