# Peer review of "The Role of Courtyards within Acute Mental Health Wards: Designing with Recovery in Mind"

_ijerph, 2022, doi:10.3390/ijerph191811414_

Round 1
Reviewer 1 Report
The purpose of this study was to obtain guidance on how exterior spaces, including courtyards, in psychiatric hospitals should be designed, based on a review of existing literature, interviews with actual inpatients of psychiatric hospitals, and an analysis of existing outstanding gardens. Beyond that, the interviews with actual inpatients (service users) in psychiatric hospitals are the most noteworthy aspect of this study, as they vividly convey the "voices of actual users" that are inaccessible to researchers and practitioners most of the time. This makes the validity of the viewpoints presented in previous studies even clearer. It is felt that the analysis of the existing excellent gardens is only informative, but would be useful to the designers. For these reasons, the study is judged to be extremely good and the paper is worthy of publication.
The following is not a particularly important point, but the word "iwi" is not familiar to me and I would appreciate some kind of explanation added if possible.
Author Response
We have reviewed the paper to improve the grammar/spelling. We have added a definition for 'iwi' with explanation. Changes can be seen in Track Changes.
Reviewer 2 Report
This article reports a very interesting and important research on the role and designing of courtyards in a recovery function within acute mental health wards. This article contributes to the topic by triangulating evidences of staff and service users and the recovery framework (CHIMES) is well-developed. The conclusions can support the performance-based model and the creation of cohesive network of micro spaces. The result seems to be scientifically sound and is interesting. The aim of the paper is quite match with the scope of the journal. However, the paper could be improved with minor corrections.
1. It is necessary to explain the recovery framework of CHIMES to readers.
2. The table3. is a mess and the comparison should be more clear and be well-connected with the elements of courtyards.
Author Response
We would like to thank this reviewer for their kind comments. We question their comment regarding the clarity of the Chimes framework. The framework is described in some detail in the Introduction (Lines 40 to 51) and then includes a description of the service implications (Line 52). Later we align the model with two leading international frameworks (Table 3) and then show how the themes relate to landscape architecture (Table 4)
We have simplified Table 3 for greater clarity. The table template for this journal did not permit the use of a grid to delineate the sections which made the links between the themes and the models/descriptions difficult. We have used shading to show which theme related to which element in the two models as well as the description. If this is not permitted we recommend removal of the table.